# Thymic Extracellular Matrix in the Thymopoiesis: Just a Supporting?

**DOI:** 10.3390/biotech11030027

**Published:** 2022-07-18

**Authors:** Marvin Paulo Lins

**Affiliations:** 1Laboratory of Cell Biology, Institute of Biological and Health Sciences, Federal University of Alagoas, Maceió 57072-970, Brazil; marvin.lins@icbs.ufal.br; 2Brazilian National Institute of Science and Technology on Neuroimmunomodulation (INCT-NIM), Oswaldo Cruz Institute, Oswaldo Cruz Foundation, Rio de Janeiro 21040-360, Brazil

**Keywords:** thymocyte, differentiation, collagen, Matrigel, ECM

## Abstract

The generation of T lymphocytes (thymopoiesis) is one of the major functions of the thymus that occurs throughout life. Thymic epithelial cells actively participate in this process. However, less attention has been paid to extracellular matrix (ECM) elements of thymus and their role in thymocyte differentiation. To clarify this topic, we selected some studies that deal with thymic ECM, its modulation, and its effects on thymopoiesis in different models. We emphasize that further studies are needed in order to deepen this knowledge and to propose new alternatives for thymic ECM functions during thymopoiesis.

## 1. The State of the Art

The extracellular matrix (ECM) is a structured and complex network composed of proteins, glycosaminoglycans, and other molecules, whose gene expression is defined as the “matrisome”. ECM is a non-cellular component present in all tissues, providing physical scaffolding for cells and biochemical cues for their differentiation and homeostasis. Similarly, it is a reservoir of several growth factors and bioactive molecules, constantly undergoing a remodeling process where its components are deposited, degraded, or modified. ECM molecules connect to cells through integrins, syndecans, and other receptors. More importantly, deregulation of ECM composition and its structure is associated with the development and progression of several physiological and pathological conditions [1,2].

In the context of the immune system, lymphoid organs exhibit significant cellularity; ECM is crucial for support and organization of the cellular microenvironments, subdividing the tissue into functional regions in order to achieve correct cell positioning. For instance, thymus has cortical and medullary regions with distinct molecular compositions at the level of the ECM. Thymic stromal cells (thymic epithelium and fibroblasts) actively produce ECM molecules, such as several types of collagens, laminins, and fibronectin [3]. Furthermore, reticulin, elastic fibers, hyaluronic acid, and heparan sulfate are present in the thymic ECM (Table 1). Expression of proteoglycans and glycosaminoglycans by thymic cells has not been extensively studied. It is known that regulatory mechanisms control thymic ECM remodeling by metalloproteinases and their tissue inhibitors [4].

Modulations in thymic ECM are observed under normal hormonal stimulation with T3, GH, prolactin, semaphorin-3A, and glucocorticoids [20]. In these situations, matrix organization may differ (in the orientation and density of proteins), although these changes are transitory and the thymus returns to its homeostatic patterns. The same does not occur under pathologic attacks from *Schistosoma mansoni*; *Trypanosoma cruzi*; *Plasmodium bergei*; cytomegalovirus infections; and diseases such as type 1 diabetes, malnutrition, myasthenia gravis, and Down’s syndrome (summarized by [21]). In these examples, the ECM network becomes larger, resulting in increased distribution and density of the molecules. The expression of chemokines is reduced, which disrupts cell positioning in the tissue. These adverse effects affect thymopoiesis, with an accumulation of thymocytes within the thymus, the early export of cells, or thymocyte death [22,23].

There is a lot of discussion about the role of thymic epithelial cells (TECs) in the differentiation of thymocytes (three-dimensional organization of TECs in the thymic microenvironment is essential for normal thymopoiesis), and participation of the ECM in this process is equally important to the function of TECs [24,25,26]. However, there is not enough data about elements of the thymic ECM as guides in the differentiation of thymocytes. In this review, we will shed light on this theme and raise new hypotheses for future investigations.

## 2. Influence of Thymic ECM in Thymopoiesis

Thymopoiesis consists of a continuous production of naïve self-tolerant T cells by the thymus to ensure an immune response against antigens and homeostasis in the peripheral tissues. In this process, T cell progenitors arise in the bone marrow from hematopoietic stem cells. These cells then migrate to the thymus and enter through vasculature at the cortical medullary junction. More immature thymocytes are double-negative since they do not express CD4 and CD8 glycoproteins. In the next stage, they become double-positive CD4^+^CD8^+^ cells. At the end of the differentiation, thymocytes become single-positive CD4^+^ or CD8^+^ cells, emigrating from the thymus [27].

During this process, thymocytes interact with thymic stromal cells through receptors and counter-receptors such as TCR-MHC, DLL4-Notch, and c-Kit-Kit ligands in the cell membranes. Furthermore, they use integrins to adhere to and receive signals from ECM components such as very late antigens (VLAs) with α and β subunits. In this case, the main obstacle in reproducing the thymic microenvironment is the complexity of its ECM composition and organization [28]. In order to support the arguments of this article, some experiments with thymic re-aggregates or artificial 3D matrices are listed hereafter. These systems, in which the ECM is preserved and modulated to a certain extent, can support limited thymopoiesis (in vitro or in vivo), though with limited success.

Bortolomai et al. (2019) [29] arranged collagen I (isolated from horses) into scaffolds such as the thymic ECM. Murine TECs survived in this system (since they reproduced a 3D environment similar to the thymic microarchitecture, perfectly recognized by TECs); however, they failed to support thymopoiesis after implantation in athymic nude mice. It is likely that the distinct homology between the species in the study (mouse and horse) made collagen unrecognizable to thymocyte receptors or changed the gene–protein expression of TECs, negatively impacting thymopoiesis.

Seach et al. (2010) [30] filled a split silicone tube (5 mm lengths and 3.35 mm internaldiameter) with Matrigel (containing laminin, collagen IV, and entactin from mouse tumor) and was engrafted with one BALB/c embryonic day-15 (E15) thymus lobe. It was then implanted around epigastric vessels in BALB/c^nu/nu^ mice. Eight weeks after grafting, T cells were found in the peripheral blood of these animals. However, there was a delayed export of these cells into the blood circulation, which may be due to a delayed time taken to vascularize the grafts, generating a lower peripheral T-cell percentage.

Given that matrix composition exhibits tissue specificity, using a native thymus matrix to reconstruct this complex organ could have an advantage over synthetic alternatives. With this in mind, Fan et al. (2015) [31] utilized bioengineered thymus organoids (from decellularized thymus scaffolds). It maintained all the major ECM components (collagen I, collagen IV, fibronectin, laminin, and glycosaminoglycans) without thymic cells. This model preserved the 3D architecture of the thymic ECM and its microstructures (grooves, ridges, and fibrillar meshwork). Thymic stromal cells and lymphocyte progenitors were co-cultivated inside these organoids and transplanted into B6 nude athymic recipients. Within 8 weeks, both CD4^+^ and CD8^+^ T cells were present in the periphery of the immune system.

Similar results were observed with human thymic cells; however, the authors did not mention participation of ECM molecules in their model [32]. In these experiments with organoids, 3D conformation of the proteins must be reviewed carefully. The isoelectric point of ECM proteins could have been better characterized, since anchorage sites for thymocytes must be natively available for correct thymopoiesis. Hun et al. (2016) [33] argued that sodium dodecyl sulfate did not preserve the general architecture of the ECM (with prominent protein denaturation), and other detergents (CHAPSO and NOG treatments) reduced fibronectin staining.

The main glycosaminoglycan-containing proteins in the thymus appeared to be damaged by all detergents (above cited) in these models, potentially impacting ECM protein and growth factor binding. Hsu et al. (2021) [34] demonstrated that the absence of heparan sulfate greatly reduced the size of the thymus in fetal thymic organ cultures and also decreased production of T cells. Among thymocyte subsets, CD4^+^ SPs were relatively increased in the absence of heparan sulfate. However, no specific blocks in T-cell development were observed. This effect can be attributed to the failure of CXCL12 and CCL19/21 chemokine immobilization and defective interstitial motility.

While the vascular framework of the ECM is well preserved in the decellularized scaffold of the thymus, significant alterations in essential factors might occur. For instance, an increase in metalloproteinases was shown to delay TCR rearrangement and inhibit the proliferation and progression of thymocytes from double-negative to double-positive, as demonstrated previously [35]. Other molecules should be investigated with these models and their effects on thymopoiesis should be studied in more detail.

It is important to highlight that, although ECM is crucial for correct thymopoiesis, other in vitro systems have been proposed and established. Trotman-Grant et al. (2021) [36] validated a model using DLL4 Fc-fusion proteins immobilized onto the surface of microbeads. Hematopoietic stem/progenitor cells (HSPCs) were cultured in this system, over seven days, which led to the emergence of T-lineage cells (CD34^−^CD7^+^CD5^+^ stage in humans). These cells were injected intrahepatically into neonatal immunodeficient NSG mice. After 12 weeks post-engraftment, there was detected mature αβTCR/CD3-bearing CD4^+^ and CD8^+^ T cells in these animals. Since this in vitro system lacks stromal elements, one of its limitations is the restricted progression to mature T cells, in addition to a probable TCR repertoire with low diversity.

## 3. Future Perspectives

From these studies, the importance of discussing thymic ECM in the context of thymopoiesis was demonstrated. There is a lot of research to be carried out, and details about the exact function of each molecule have to be described. Ligands and receptors, stromal cells, and thymocytes interact with each other and with the thymic ECM. More attention should be paid to thymic ECM, since correct tissue orientation and functionality depend on it.

Another topic of future discussion is the use of human organoid cultures, with several potential benefits, such as faster and more robust outcomes, and a more accurate representation of human tissue, such as thymus, in order to generate immunocompetent T cells.

In the future, adjuvant immunotherapies (such as thymic rejuvenation to increase T cell export) might be developed for immunodeficient patients or autoimmune diseases if the thymic ECM could be seen as a potential topic for further studies.

## Figures and Tables

**Table 1 biotech-11-00027-t001:** Major extracellular matrix components of thymus.

ECM Component	Localization	References
Elastic fibers	Capsule	[5]
Entactin	Cortex	[6]
Fibronectin	Basement membranes, Thymic septa, Perivascular space, and Cortex and medulla	[7,8]
Galectin-3	Corticomedullary junction and medulla	[9]
Heparan sulfate	Medulla and blood vessels	[10]
Hialuronic acid	Capsule, septum and vessels	[11]
Laminin	Basement membranes, Thymic septa, Perivascular space, Cortex, and medulla	[7,8,12]
Nidogen	Capsule, cortex, and medulla	[13]
Perlecan	Blood vessels and capsule	[14]
Tenascin	Corticomedullary junction, Medulla	[15]
Type I collagen	Basement membrane, Septum, and blood vessels	[16]
Type III collagen (reticulin)	Basement membrane, Subjacent to capsule, Septum, and blood vessels	[16,17]
Type IV collagen	Basement membranes, Thymic septa and Perivascular space	[7]
Versican	Minimal expression	[18]
Vitronectin	Death thymocytes	[19]

## Data Availability

Not applicable.

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
