# Peer review of "Thymic Extracellular Matrix in the Thymopoiesis: Just a Supporting?"

_biotech, 2022, doi:10.3390/biotech11030027_

Round 1

Reviewer 1 Report

The authors have written a nice correspondence about a critical but still highly overlooked subject within thymus function and regeneration. The authors make their point that extracellular matrix in the thymus is important, by stating the results of some interesting in vivo model studies reporting on experiments with thymic re-aggregates

or artificial 3-D matrices. This manuscript is highly interesting and well-written, but can improve with some adjustments.

Comment 1: the authors build their case that the thymic ECM is complex and highly important for correct thymopoiesis by stating only the results of studies reporting on experiments with thymic re-aggregates or artificial 3-D matrices. While these are interesting results, the authors should consider also touching upon results from recent studies reporting on thymopoiesis after progenitor T cell infusion in mice (a.o. work from the lab of Zuniga-Pflucker); in which mice with irradiated thymi were infused with CD5+CD7+ proT cells.

Comment 2: in addition to comment 1, the authors should comment on the results from recent reports that applied in vitro notch signalling (OP9-DL1, OP9-DL4, beads, etc). As these reports also show potent T-cell development from stem cells: what does that say about the importance of thymic ECM?

Comment 3: If possible, please provide references for lines 29-34:  In the context of the immune system, lymphoid organs exhibit significant cellularity ECM is crucial for support and organization of the cellular microenvironment, subdividing the tissue into functional regions, aiming at correct cell positioning. For instance, thymus has cortical and medullary regions with distinct molecular compositions at the level of the ECM. Thymic stromal cells (thymic epithelium and fibroblasts) actively produce ECM molecules, such as several types of collagens, laminin, and fibronectin.

Comment 4: If possible, please provide references for lines 66-70: During this process, thymocytes interact with thymic stromal cells through receptors and counter-receptors such as TCR-MHC, Dll4-Notch, and c-Kit-Kit ligand in their cell membranes. Furthermore, they use integrins to adhere to and receive signals from ECM components such as VLAs, with α and β subunits. In this case, the main obstacle in reproducing the thymic microenvironment is the complexity of its ECM composition and organization.

Comment 5: In line 71, the authors write: “in order to support our discussion…”. However, it is unclear to the reader what this discussion is at this point. Please clarify.

Comment 6: in future perspectives, the authors state that “In the near future, adjuvant immunotherapies can be developed for immunodeficient patients or autoimmune diseases…”. Although this reviewer agrees that targeting thymic ECM has clinical relevance, it is 1) not clear what kind of immunotherapies the authors mean; please clarify, and 2) that this would be in the “near” future; please consider to leave out the “near” in this sentence.

Author Response

Comment 1: the authors build their case that the thymic ECM is complex and highly important for correct thymopoiesis by stating only the results of studies reporting on experiments with thymic re-aggregates or artificial 3-D matrices. While these are interesting results, the authors should consider also touching upon results from recent studies reporting on thymopoiesis after progenitor T cell infusion in mice (a.o. work from the lab of Zuniga-Pflucker); in which mice with irradiated thymi were infused with CD5+CD7+ proT cells.

I am very grateful for this new look at the manuscript, and I followed this orientation: I added in the text the article that the reviewer cited

Comment 2: in addition to comment 1, the authors should comment on the results from recent reports that applied in vitro notch signalling (OP9-DL1, OP9-DL4, beads, etc). As these reports also show potent T-cell development from stem cells: what does that say about the importance of thymic ECM?

This comment is in line with the previous one and shows that, in the early stages of thymopoiesis, DL4-Notch is an indispensable interaction for thymocytes via TECs. However, in the later stages, the presence of the thymic ECM is crucial for this development, as the same authors of the article above discuss about the limitations of these model.

Comment 3: If possible, please provide references for lines 29-34:  In the context of the immune system, lymphoid organs exhibit significant cellularity ECM is crucial for support and organization of the cellular microenvironment, subdividing the tissue into functional regions, aiming at correct cell positioning. For instance, thymus has cortical and medullary regions with distinct molecular compositions at the level of the ECM. Thymic stromal cells (thymic epithelium and fibroblasts) actively produce ECM molecules, such as several types of collagens, laminin, and fibronectin.

This note was important. I added the reference to the manuscript:

Thapa P, Farber DL. The Role of the Thymus in the Immune Response. Thorac Surg Clin. 2019 May;29(2):123-131. doi: 10.1016/j.thorsurg.2018.12.001. Epub 2019 Mar 7. PMID: 30927993; PMCID: PMC6446584.

Comment 4: If possible, please provide references for lines 66-70: During this process, thymocytes interact with thymic stromal cells through receptors and counter-receptors such as TCR-MHC, Dll4-Notch, and c-Kit-Kit ligand in their cell membranes. Furthermore, they use integrins to adhere to and receive signals from ECM components such as VLAs, with α and β subunits. In this case, the main obstacle in reproducing the thymic microenvironment is the complexity of its ECM composition and organization.

As per the reviewer's guidance, I have added a reference to this part of the manuscript.

Asnaghi MA, Barthlott T, Gullotta F, Strusi V, Amovilli A, Hafen K, Srivastava G, Oertle P, Toni R, Wendt D, Holländer GA, Martin I. Thymus Extracellular Matrix-Derived Scaffolds Support Graft-Resident Thymopoiesis and Long-Term In Vitro Culture of Adult Thymic Epithelial Cells. Adv Funct Mater. 2021 May 17;31(20):2010747. doi: 10.1002/adfm.202010747. Epub 2021 Mar 12. PMID: 34539304; PMCID: PMC8436951.

Comment 5: In line 71, the authors write: “in order to support our discussion…”. However, it is unclear to the reader what this discussion is at this point. Please clarify.

I appreciate this suggestion. I made a substitution to make the text clearer to the reader.

Comment 6: in future perspectives, the authors state that “In the near future, adjuvant immunotherapies can be developed for immunodeficient patients or autoimmune diseases…”. Although this reviewer agrees that targeting thymic ECM has clinical relevance, it is 1) not clear what kind of immunotherapies the authors mean; please clarify, and 2) that this would be in the “near” future; please consider to leave out the “near” in this sentence.

This comment makes a lot of sense. I explained what type of immunotherapy I referred to in the manuscript and removed the word "near"

Reviewer 2 Report

This is an important topic and the author Marvin Paulo Lins did complete justice to this subject. However, I believe author should discuss a little bit the role of ECM in neurodegenerative diseases such as Alzheimer’s disease where extracellular amyloid beta protein propagates in a prion like manner.

As 3D culture allows to visualize complex network interactions, it will be very important to note the advantages of organoid based models in the future perspective section a bit more detail.

Author Response

This is an important topic and the author Marvin Paulo Lins did complete justice to this subject. However, I believe author should discuss a little bit the role of ECM in neurodegenerative diseases such as Alzheimer’s disease where extracellular amyloid beta protein propagates in a prion like manner.

I would like to thank you for your appreciation of my manuscript and the reviewer's suggestions. And, although I consider neurodegenerative diseases is an important topic to be discussed (regarding ECM), this topic deviates a little from my initial proposal: the role played by ECM regarding thymopoiesis, its modulations and perspectives.

As 3D culture allows to visualize complex network interactions, it will be very important to note the advantages of organoid based models in the future perspective section a bit more detail.

This is an excellent suggestion. I added this subject to the “Future Perspectives” topic in my manuscript.

Reviewer 3 Report

The viewpoints are clear. Some references should be included, and some spelling issues should be paid attention to. Please find the details highlighted in the attached PDF. 

Author Response

The viewpoints are clear. Some references should be included, and some spelling issues should be paid attention to. Please find the details highlighted in the attached PDF. 

I would like to thank you for the notes made in the manuscript. I followed all the suggestions recommended in the final text.

Round 2

Reviewer 1 Report

I have no further comments. Great paper.